# Progressive Multi-scale Triplane Network for Text-to-3D Generation

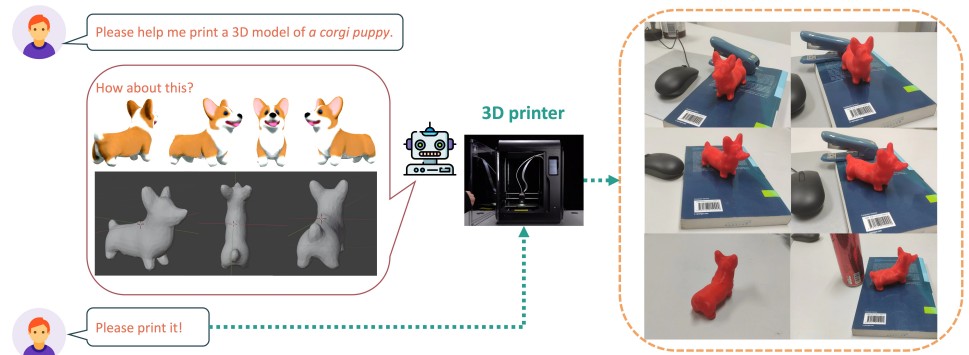

Figure 1: The proposed algorithm facilitates effortless and interactive creation of high-quality 3D meshes from natural language descriptions, which can then be utilized for 3D printing. The six images at the right show the corresponding physical 3D printed model from multiple perspectives. Our output meshes are ready for 3D printing. (We add the book and mouse as the size reference.)

## Abstract

The challenge of text-to-3D generation lies in accurately and efficiently crafting 3D objects based on natural language descriptions, a capability that promises substantial reduction in manual design efforts and offers an intuitive interface for user interaction with digital environments. Despite recent advancements, effective recovery of fine-grained details and efficient optimization of high-resolution 3D outputs remain critical hurdles. Drawing inspiration from the efficacious paradigm of progressive learning, we present a novel Multi-scale Triplane Network (MTN) architecture coupled with a tailored progressive learning strategy. As the name implies, the Multi-scale Triplane Network consists of four triplanes transitioning from low to high resolution. This hierarchical structure allows the low-resolution triplane to serve as an initial shape for the high-resolution counterparts, easing the inherent complexity of the optimization process. Furthermore, we introduce the progressive learning scheme that systematically guides the network to shift its attention from prominent coarse-grained structures to intricate fine-grained patterns. This strategic progression ensures that the focus of the model evolves towards emulating the subtlest aspects of the described 3D object. Our experiment verifies that the proposed method performs favorably against contemporary methods. Even for the complex and nuanced textual descriptions, our method consistently excels, delivering robust and viable 3D shapes where other methods falter.

## 1 Introduction

Designing digital models for manufacturing (Drotman et al., 2017; Fujii et al., 2023) is often time-consuming and labor-intensive. To streamline this process, researchers are developing more intuitive methods for 3D object generation, such as using text prompts (see Figure 10). The aim of the text-to-3D generation task is to automatically create a 3D object draft from a natural description, thus cutting down the design efforts from the ground up.

In recent years, text-to-3D generation has reported rapid development due to the breakthrough of text-to-image diffusion models (Dhariwal & Nichol, 2021; Nichol & Dhariwal, 2021; Song et al.,

*"a corgi puppy"*    *"a tiger cub"*    *"a rabbit, animated movie character, high detail 3d model"*    *"a bust of a mouse"*    *"a toy motorcycle"*    *"a cake in the shape of a train"*

Figure 2: Our method is able to generate high-quality 3D outputs from various text prompts using the proposed Multi-scale Triplane Network (MTN). We display both mesh normals and the generated results obtained from texts of varying lengths. Specifically, our approach showcases the ability to create animal meshes and industrial products. Moreover, automatic color rendering is applied when a common color is applicable for such a category.

2021). For instance, the pioneer work DreamFusion (Poole et al., 2023) leverages the 2D Stable Diffusion and proposes Score Distillation Sampling (SDS) algorithm to generate a variety of 3D objects using only text prompts. However, there remain two problems: **1)** The inherent optimization complexity of 3D high-resolution objects. It is hard to directly map one sentence to one high-dimension 3D object, especially in the form of Neural Radiance Fields (NeRF) (Mildenhall et al., 2021). This leads to either generation collapse or extended training duration for model convergence. **2)** Lack of fine-grained details. We notice that some works report blurred results (Poole et al., 2023; Wang et al., 2023a; Metzer et al., 2023). This is due to the use of a fixed training strategy, *i.e.*, focusing on global fidelity all the time while ignoring local parts.

In an attempt to overcome the above-mentioned challenges, we propose a progressive text-to-3D generation model that can gradually refine details to produce high-quality 3D objects (see Figure 2). **1)** For the first problem, we introduce a novel network structure, namely, Multi-scale Triplane Network (MTN) consisting of four triplanes ranging from low to high resolution. In the initial phases of training, we sample low-resolution features from the corresponding low-resolution triplane to capture the basic global geometric shape. As training advances, we fix the former low-resolution triplanes and gradually shift our focus to triplanes with a higher resolution. Such a progressive structure facilitates the model to capture different-level features in a step-by-step manner and thus enhances the geometric and textural nuances of the 3D model, such as color and texture. **2)** For the second problem, we adopt a progressive learning strategy focusing on two key factors, *i.e.*, time step $t$ and camera radius. In particular, unlike existing 2D diffusion models that utilize random sampling, we adopt a large $t$ during the initial stages to guide the global structure. As the training progresses, we transition to a smaller $t$ to refine visual details. Meanwhile, we gradually adjust the radius of the camera to approach the object more closely. This enables the camera to initially focus on capturing the global structure and later shift its attention to the local details. Our contributions are as follows:

- We introduce a Multi-scale Triplane Network (MTN) to effectively tackles the challenge of text-to-3D generation in a bottom-up manner. This hierarchical structure progresses from rough to fine-grained details, leveraging initial low-resolution shapes to streamline the high-resolution optimization, overcoming complexities faced by prior methods.

- We propose a progressive learning strategy tailored for the Multi-scale Triplane Network. It simultaneously reduces the camera radius and time step $t$ in diffusion to refine details of the 3D model in a coarse-to-fine manner, ensuring superior capture of subtle details in the generated 3D models.

- Albeit simple, extensive experiments show that the proposed method could achieve high-resolution outputs that align closely with natural language descriptions. We expect this work to pave the way for automatic 3D printing via intuitive human-machine interaction.

## 2 RELATED WORK

**3D Generative Modeling** The realm of 3D generative modeling has seen extensive exploration across diverse representation types, including voxel grids (Tatarchenko et al., 2017; Li et al., 2017), point clouds (Luo et al., 2021; Zhou et al., 2021; Vahdat et al., 2022), meshes (Gao et al., 2019; 2021; Nash et al., 2020; Henderson et al., 2020; Gupta, 2020; Rosinol et al., 2019), implicit fields (Cheng et al., 2022; Wu et al., 2020; Wu & Zheng, 2022; Zheng et al., 2022a), and octrees (Ibing et al.,

2023). While many traditional approaches hinge on 3D assets as training data, the challenge of acquiring such data at scale has spurred alternative strategies. Addressing the inherent challenge of obtaining 3D assets for training, some recent endeavors have turned to 2D supervision. Leveraging ubiquitous 2D images, models, *e.g.*, pi-GAN (Chan et al., 2021), EG3D (Chan et al., 2022), MagicMirror (Zheng et al., 2022b), and GIRAFFE (Niemeyer & Geiger, 2021) have supervised 2D renderings of 3D models through adversarial loss against 2D image datasets. While these approaches hold potential, a recurring challenge is that they are often restricted to specific domains, *e.g.*, human faces (Karras et al., 2019), limiting their versatility and hindering expansive creative freedom in 3D design. In our study, we pivot towards text-to-3D generation, with the goal of crafting visually favorable 3D objects guided by diverse text prompts.

**Text-to-3D Generation** The success of text-to-image generation models has driven substantial progress in the emerging field of text-to-3D object generation. Notably, the integration of CLIP into models, *e.g.*, CLIP-forge (Sanghi et al., 2022), Dream Fields (Jain et al., 2022), Text2Mesh (Michel et al., 2022), CLIPmesh (Mohammad Khalid et al., 2022), and CLIP-NeRF (Wang et al., 2022) has been a significant advancement. These approaches harness CLIP to optimize 3D representations, ensuring that 2D renderings resonate with textual prompts. A defining advantage of such techniques is their ability to bypass the need for costly 3D training data, though a trade-off in terms of the realism of the resultant 3D models has been observed. More recent advancements, such as DreamFusion (Poole et al., 2023), which proposes Score Distillation Sampling (SDS) Loss, SJC (Wang et al., 2023a), Magic3D (Lin et al., 2023), and Latent-NeRF (Metzer et al., 2023), have showcased the merits of employing robust text-to-image diffusion models as a robust 2D prior, elevating the quality and realism of text-to-3D generation. Such a visual prior, capitalizing on the potential of diffusion models, has led to outcomes with higher fidelity and diversity, as well as reduced generation time. Along this line, Fantasia3D (Chen et al., 2023) employs disentangled modeling of geometry and appearance, enhancing fidelity and realism while offering better control over both properties. Meanwhile, ProlificDreamer (Wang et al., 2023b) introduces Variational Score Distillation (VSD) Loss, serving as a replacement for SDS Loss. This enhancement has resulted in outputs characterized by higher resolution and increased diversity in 3D representations. Despite these advances, the multi-face (Janus) problem remains. To address this, Zero-1-to-3 (Liu et al., 2023), Image-Dream (Wang & Shi, 2023), and MVDream (Shi et al., 2024) introduce multi-view diffusion models trained on extensive multi-view datasets to ensure multi-view consistency. Additionally, Bidiff (Ding et al., 2024) presents a unified framework integrating 3D and 2D diffusion processes to preserve both 3D fidelity and 2D texture richness. While substantial, these contributions differ from our focus on enhancing 3D representation quality and can complement our method. Triplane-based methods, such as Instant3D (Li et al., 2023), DIRECT-3D (Liu et al., 2024), and TPA3D (Wu et al., 2025) represent a promising alternative within the NeRF-based text-to-3D landscape. By leveraging efficient Triplane representations, these approaches achieve a balance between computational efficiency and output quality. These methods reveal the potential of Triplane representations to elevate text-to-3D generation tasks and align closely with the principles of our approach. Recently, 3D Gaussian Splatting (Kerbl et al., 2023) has emerged as an alternative to NeRF. Methods like DreamGaussian (Tang et al., 2024), GSGEN (Chen et al., 2024), GaussianDreamer (Yi et al., 2024), and LucidDreamer (Liang et al., 2024) have applied this representation to text-to-3D generation. Though faster, these approaches often compromise the high quality characteristic of NeRF-based methods and require post-processing to convert Gaussian representations into NeRF or meshes, adding computational overhead. Therefore, we focus on NeRF-based methods for their superior quality and fidelity, building upon the principles of this line of research and introduce novel techniques to effectively enhance the quality of 3D outputs.

## 3 METHOD

### 3.1 MULTI-SCALE TRIPLANE

An overview of our Multi-scale Triplane Network (MTN) is shown in Figure 3 (a). In particular, MTN is composed of four triplanes (Chan et al., 2022) ranging from low to high resolutions. Each triplane leverages three axis-aligned 2D feature planes $\mathbf{F}_{xy}^m, \mathbf{F}_{xz}^m, \mathbf{F}_{yz}^m \in \mathbb{R}^{N_m \times N_m \times C}, m = 1, 2, 3$. $N_m$ denotes spatial resolution, while $C$ is the dimension of the channels and $m$ represents the training stage. Note that a large $N_m$ results in a substantial GPU memory cost. Therefore, for the last triplane, we essentially employ a trivector instead to optimize memory usage and support higher

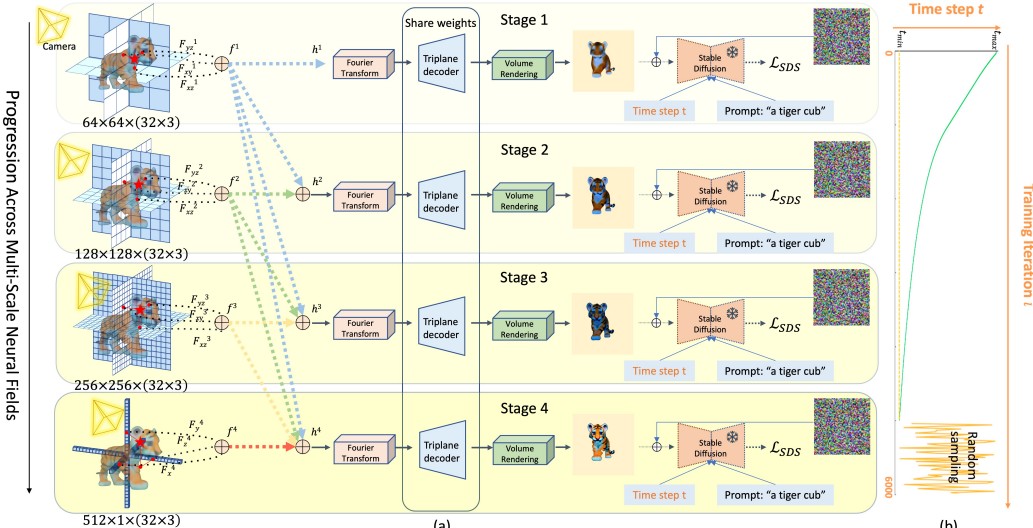

Figure 3: Overview of the proposed Multi-scale Triplane Network (MTN). **(a)** Given the text prompts, *e.g.*, "a tiger cub", MTN generates 3D representations using Multi-scale Neural Fields, utilizing four triplanes varying in resolution. To save memory costs and enable the highest resolution, we make a trade-off to deploy the high-dimension trivector format as the triplane alternative. First, by casting rays from a random camera position and view, we can sample a lot of 3D points along each ray and then encode their corresponding features by projecting them onto triplanes. After the 3D input encoding, the network uses a Fourier transform, a triplane decoder, and volume rendering. The Fourier feature transform (Tancik et al., 2020) enables the triplane decoder to learn high-frequency information. The network employs Fourier transform, a shallow MLP triplane decoder, and volume rendering to convert the 3D representation into RGB images. Training progresses in four stages, starting with low-resolution triplanes for global geometric insights, and gradually shifting to higher-resolution triplanes for detailed refinement. **(b)** Concurrently, as training proceeds, the time step $t$ undergoes progressive adjustments, and the camera also approaches the neural field progressively, emphasizing the refinement of local features. To update the parameters, we employ a frozen Stable Diffusion model to estimate the injected noise on the rendered image (*e.g.*, tiger) and then backpropagate the gradient.

resolution. This trivector configuration leverages the axis-aligned vectors $\mathbf{F}_x^4, \mathbf{F}_y^4, \mathbf{F}_z^4 \in \mathbb{R}^{N_4 \times 1 \times C}$ with a resolution of $N_4 \times 1$ and $C$.

Given any 3D coordinate point $p \in \mathbb{R}^3$, we project this coordinate onto each of these orthogonal feature planes and sample feature vectors via interpolation. We then sum these three vectors $f^m(p) = \mathbf{F}_{xy}^m(p) + \mathbf{F}_{xz}^m(p) + \mathbf{F}_{yz}^m(p)$ for $m = 1, 2, 3$ as position features for the first three triplanes, while $f^4(p) = \mathbf{F}_x^4(p) + \mathbf{F}_y^4(p) + \mathbf{F}_z^4(p)$ for the last trivector. To aggregate multi-scale features, we further fuse the different level position features together as $h^m(p) = \sum_{k=1}^m (f^k(p))$. After obtaining the multi-scale representation, we follow Tancik et al. (2020) to transform the summed position features into the Fourier domain. Subsequently, the Fourier features are fed forward into a lightweight triplane decoder to estimate color and density (Mildenhall et al., 2021). We deploy a Multi-Layer Perceptron (MLP) as the triplane decoder. Finally, to calculate the loss, we apply neural volume rendering techniques (Mildenhall et al., 2021) to project the 3D representation onto an RGB image $I$, which is the input of the Diffusion model.

**Discussion. Why is a multi-scale structure crucial?** As shown in Figure 3, we apply triplanes with different resolutions to capture features at multiple scales. This approach is designed to mimic the human recognition system, which transitions from recognizing basic elements to more intricate details when observing 3D objects. For example, when a person sees a new object, they first perceive its overarching structure and then refine the details through foveal vision. During the early stages of training, we extract low-resolution features from the corresponding low-resolution triplane. Each point on the low-resolution triplane, obtained through interpolation from a coarse grid, encompasses a broader field of view, providing global geometric insights. As training progresses, we gradually shift our focus to higher-resolution triplanes, which can capture intricate features and refine details

such as subtle shading and texture nuances. This process facilitates the optimization of high-scale features, especially when low-scale features have already been well-optimized. This multi-scale approach is conceptually similar to curriculum learning (Bengio et al., 2009), where the model starts with simpler tasks and gradually advances to more complex ones. In the experiments, we observe that the proposed method achieves visual enhancements in both shape and texture of the model, even for complex descriptions.

**Optimization objective.** Given the projected image $I$, we apply Score Distillation Sampling (SDS) (Poole et al., 2023) to distill 2D image priors from the pretrained 2D diffusion model $\epsilon_\phi$. The loss on 2D projection is then back-propagated to update differentiable 3D representations. In particular, the proposed 3D model can be typically depicted as a parametric function $I = g_\theta(P)$, where $I$ represents the images produced at distinct camera poses, and $P$ is the set of multiple positions $p$. Here, $g$ denotes the volumetric rendering mechanism, and $\theta$ embodies a coordinate-based MLP and triplanes that portray a 3D scene. To estimate the projection quality, we adopt the pretrained diffusion model, which is well aligned with text prompts $y$. The one-time denoising forward can be formulated as $\epsilon_\phi(I_t; y, t)$ to predict the noise $\varepsilon$ given the noisy image $I_t$, time step $t$, and text embedding $y$. Therefore, the gradient of the SDS loss can be formulated as:

$$\nabla_\theta \mathcal{L}_{SDS}(\phi, g_\theta(P)) = \mathbb{E}_{t,\epsilon} \left[ (\epsilon_\phi (I_t; y, t) - \epsilon) \frac{\partial I_t}{\partial \theta} \right],$$

where $\epsilon$ is a noise term following a standard normal distribution and $I_t$ denotes the noisy image. Following the setting in the diffusion model (Dhariwal & Nichol, 2021; Nichol & Dhariwal, 2021; Song et al., 2021), the noisy image can be formulated as a linear process $I_t = \sqrt{\bar{\alpha}_t} I + \sqrt{1 - \bar{\alpha}_t} \epsilon$, where $\bar{\alpha}_t$ is a predefined time-dependent constant. Besides, it is worth noting that the diffusion model parameter $\phi$ is frozen. The purpose of this denoising function is to offer the text-aware guidance to update $\theta$. If the projection $I$ is well-aligned with the text $y$, the noise on $I_t$ is easy to predict. Otherwise, we will punish the 3D model.

## 3.2 Progressive Learning Strategy

Another essential element underlying the proposed method is the employment of a progressive learning strategy, focusing on two critical parameters, *i.e.*, the time step $t$ and camera radius.

**Progressive time step sampling.** We first introduce a progressive time step ($t$) sampling approach. It is motivated by the observation that the default uniform $t$-sampling in SDS training often results in inefficiencies and inaccuracies due to the broad-range random sampling. Our approach, therefore, emphasizes a gradual reduction of the time step, directing the model to transition from coarse to detailed learning (See Figure 3 (b)). In the early phases of training, we adopt larger time steps to add a substantial amount of noise into the image. During the noise recovery process, the network is driven to focus on the low-frequency global structure signal. As training advances and the global structure stabilizes, we decrease to smaller time steps. In this stage, the network is demanded to recover the high-frequency fine-grained pattern according to the context. It facilitates the model in refining local details, such as textures and shades. We define the rate of change of variable $t$ as:

$$\frac{\mathrm{d}t}{\mathrm{d}i} = \beta v(t), \tag{1}$$

where $v(t)$ controls how $t$ changes with respect to the training iteration $i$ and is manually designed. $\beta$ is a positive constant. We define $v(t)$ piece-wise:

$$v(t) = \begin{cases} -\exp(\frac{t - n_2}{m_2}) & \text{if } t > n_2 \\ -1.0 & \text{if } n_1 \le t \le n_2 \\ -\exp(\frac{t - n_1}{m_1}) & \text{if } t < n_1, \end{cases} \tag{2}$$

Here, $v(t) < 0$ implies $\frac{\mathrm{d}t}{\mathrm{d}i} < 0$, indicating that $t$ decreases as training progresses. Our design ensures that $t$ decreases rapidly at the beginning ($t > n_2$), linearly in the middle ($n_1 \le t \le n_2$), and more mildly towards the end ($t < n_1$). After the time step $t$ decreases to $t_{\min}$, we revert to random sampling from a uniform distribution as: $t \sim \mathcal{U}(t_{\min}, t_{\max})$, where $\mathcal{U}(t_{\min}, t_{\max})$ denotes uniform sampling within the interval from $t_{\min}$ to $t_{\max}$. It reintroduces randomness to maintain the vibrancy of the coloration of the 3D model. We notice that a concurrent work, Dreamtime (Huang et al., 2024),

also employs a similar non-increasing $t$-sampling strategy. However, such a strategy sometimes tends to overfit the local details, and inadvertently change the global illumination. Therefore, it is crucial to avoid the consistent use of extremely small time steps at the end of training. Different from Dreamtime (Huang et al., 2024), our method decreases $t$ with the training step at a much steeper pace and employs a mixture of both deterministic and random sampling as shown in Figure 3 (b).

**Progressive radius.** Simultaneously, our approach also incorporates a dynamic camera radius considering the camera movements in the real world. Typically, eyes will move closer for detailed object observation. Motivated by this behavior, we dynamically adjust the camera radius during the multi-scale learning. During the low-scale triplane stage, which focuses on broader geometric structures, we utilize a large camera radius to cover the entire object. As we move to the high-scale triplane stage, which refines local model details, the camera radius is reduced to closely focus on finer details of the 3D scene. This progressive radius strategy is intuitive and directly impacts resolution, aiding in feature learning across varying scales. In the ablation study, we also verify the effectiveness of this strategy (See Section 4.3).

### 3.3 IMPLEMENTATION DETAILS

**Neural field rendering structure.** The proposed MTN consists of three triplanes and one trivector varying in resolution. The resolutions of the triplanes $N_1, N_2, N_3 = 64, 128, 256$, and the number of channels $C = 32$. For the trivector, we set $N_4 = 512$. During the Neural Field optimization, camera positions are randomly sampled in spherical coordinates. The azimuth angles, polar angles and fovy range are randomly sampled between $[-180°, 180°]$, $[45°, 105°]$, and $[10°, 30°]$, respectively. For spherical radius of the camera, the initial $R \in [3.0, 3.5]$ and gradually decreases to $R \in [1.8, 2.1]$.

**Prompts.** For prompt augmentation, the default view-dependent prompt augmentation appends corresponding view, *e.g.*, " front view", "back view", and "side view" according to the camera position. However, we adopt the strategy from Perp-Neg (Armandpour et al., 2023), leveraging geometric properties to enhance the diffusion model's alignment with user prompts. This approach enriches original prompts with view-dependent conditional text embeddings based on sampled camera positions, ensuring the rendered image adheres to the desired view. Specifically, if the azimuth angle $\phi \in [-90°, 90°]$, a soft embedding is interpolated between "front view" and "side view" based on $\phi$ and appended to the original text embedding. Conversely, for $\phi \notin [-90°, 90°]$, the algorithm interpolates between "back view" and "side view" embeddings. This nuanced addition ensures more accurate and user-aligned renderings.

**Diffusion model.** We deploy DeepFloyd-IF (Konstantinov, 2023) as the guidance model to provide 2D image priors. For time step $(t)$ sampling in SDS, the Stable-DreamFusion uses random sampling $t \sim \mathcal{U}(20, 980)$. In our proposed approach, the time step $t$ is set to decrease from 980 to 20. Through a grid search, we empirically set an optimal prior weight configuration as $\{m_1 = 50, m_2 = 150, n_1 = 500, n_2 = 800\}$ to control the rate of decrease. Following existing works (Poole et al., 2023; Lin et al., 2023; Armandpour et al., 2023), we also adopt the viewpoint-aware prompts by appending prompts such as "front view", "side view", and "back view".

**Optimization.** The number of total iterations is 6000 and the batch size is 1. We employ the Adan optimizer (Xie et al., 2022) with learning rate of $1 \times 10^{-3}$, weight decay of $2 \times 10^{-5}$. Following the existing work (Chan et al., 2022), we apply two regularization terms, *i.e.*, TV regularization and L2 regularization, to prevent floating clouds. The model can converge within one hour on a V100 GPU. Specifically, we configure the training process with 3,000 iterations for the first stage, followed by 1,000 iterations each for the second, third, and final stage, respectively.

## 4 EXPERIMENT

In this section, we assess the capability of our method to produce high-fidelity 3D objects according to natural language prompts. We primarily consider three key evaluation aspects: (1) alignment with the text, particularly focusing on key words in the sentence; (2) intricate texture details; and (3) consistent geometric shape, especially in localized parts, *e.g.*, ears and tails. Due to the space limitation, we mainly compare our approach against four widely-used text-to-3D frameworks. Since DreamFusion (Poole et al., 2023) is not publicly available, we utilize the open-source variant, Stable-DreamFusion (Tang, 2022). Besides, we also compare the proposed method with other three

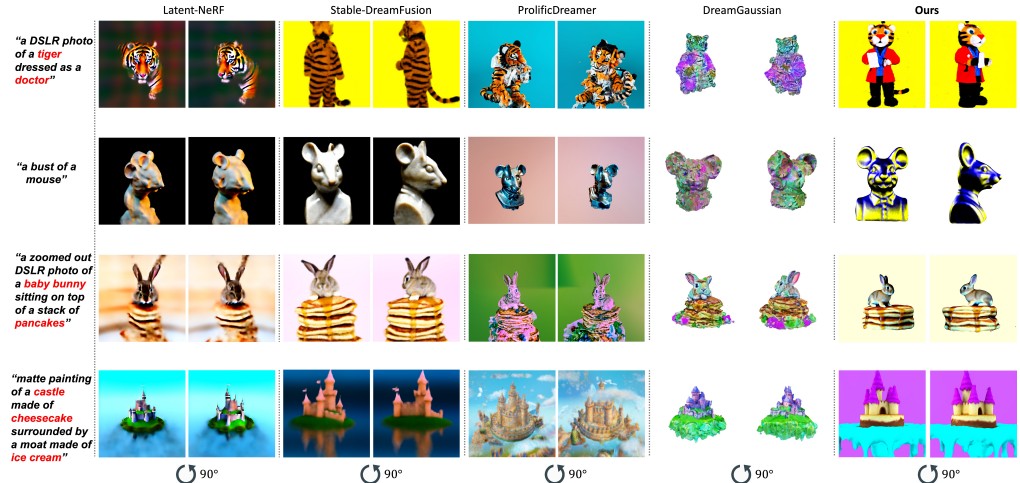

Figure 4: Qualitative comparisons for text-to-3D generation among our method, Latent-NeRF (Metzer et al., 2023), Stable-DreamFusion (Tang, 2022), ProlificDreamer (Wang et al., 2023b), and DreamGaussian (Tang et al., 2024). Here we show the 2D projection of the front view and side view of the 3D model. We observe that the proposed method could generate a higher-fidelity 3D representation aligned with the given description, reducing the extra post-processing costs. **In the last row, despite ProlificDreamer (Wang et al., 2023b) and Latent-NeRF (Metzer et al., 2023) achieves good visual quality, they generally miss the keyword "cheesecake" and "ice cream".**

competitive works, *i.e.*, Latent-NeRF (Metzer et al., 2023), ProlificDreamer (Wang et al., 2023b), and DreamGaussian (Tang et al., 2024).

## 4.1 QUALITATIVE EVALUATION

As shown in Figure 4, we could observe that our method outperforms prior competitive approaches in terms of text alignment, texture details, and geometric precision. The qualitative analysis reveals the superior performance of our method in generating realistic and accurate 3D representations aligned with textual prompts. In the first row, we observe notable deficiencies in Latent-NeRF (Metzer et al., 2023), which struggles to produce a coherent 3D model. While Stable-DreamFusion (Tang, 2022) manages to generate a tiger avatar, it fails to incorporate the crucial keyword "doctor". ProlificDreamer (Wang et al., 2023b), despite its high output resolution, erroneously includes unrelated elements, such as a camera, on the tiger's hand, which is obviously inconsistent with the specified theme of "a tiger doctor." DreamGaussian (Tang et al., 2024), on the other hand, successfully identifies the "tiger face" element but falters in rendering the rest of the model, resulting in an overall geometry that appears unconventional. In contrast, our proposed method seamlessly integrates the textual cues to craft a detailed representation of a tiger doctor, complete with a book in its hands. In the second row, our method presents a refined geometric shape with correct shading on the bust, surpassing Stable-DreamFusion (Tang, 2022), which erroneously places a tail on the head. Similarly, the outputs from Latent-NeRF (Metzer et al., 2023), ProlificDreamer (Wang et al., 2023b), and DreamGaussian (Tang et al., 2024) display inaccuracies in head shape, notably featuring three ears and multi-face. Additionally, DreamGaussian (Tang et al., 2024) shows discrepancies in color saturation, resulting in outputs that are excessively vibrant. Simultaneously, our method distinguishes itself by depicting nuanced features such as the necktie and buttons on the mouse. In the third row, our method accurately captures the keyword "baby bunny", showcasing a natural geometric shape with clear edges and appropriate features. Conversely, both Latent-NeRF (Metzer et al., 2023) and Stable-DreamFusion (Tang, 2022) continue to struggle with the multi-face and multi-ear issue. ProlificDreamer (Wang et al., 2023b), and DreamGaussian (Tang et al., 2024), while offering high-resolution outputs, fall short in aligning their geometric shapes and color fidelity with the textual prompt, underscoring the critical balance between resolution and semantic coherence. In the last row, our method aligns well with the given text prompt, accurately capturing the three keywords "castle", "cheesecake", and "ice cream", and generates high-quality 3D outputs with exquisite textures. In contrast, other methods primarily focus on the keyword "castle" and overlook the additional critical details. Although ProlificDreamer (Wang et al., 2023b) produces a visually appealing scene

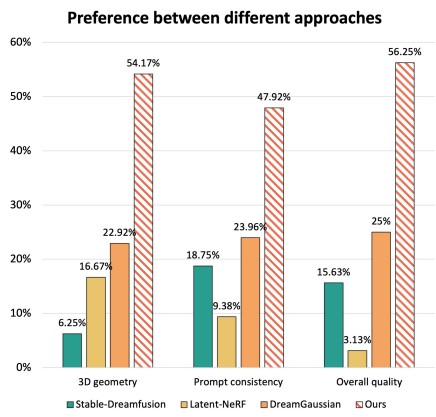

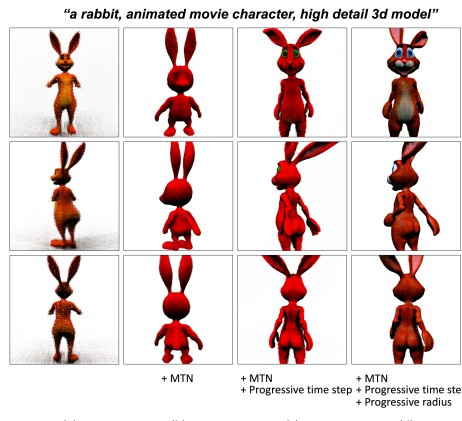

Figure 5: User study on visual quality. The proposed method excels in 3D geometry, closely aligns with user prompts, and outperforms two competitive approaches in overall quality.

Figure 6: Ablation study of the primary components. (a) Single triplane; (b) add MTN architecture; (c) add Progressive time step strategy; (d) add Progressive radius, which is our full method. Our full model crafts a delicate geometric shape and achieves accurate texture.

with diverse features, its output appears foggy and cloud-filled, which deviates noticeably from the given prompt. In summary, our method excels in producing reliable and precise 3D models that align seamlessly with textual prompts, reflecting naturally intuitive geometric shapes that resonate well with human intuition.

**User Study.** For a more comprehensive evaluation, we conduct a user study with 96 participants. We evaluate our model against three prevailing and basic approaches, *e.g.*, Latent-NeRF (Metzer et al., 2023), Stable-DreamFusion (Tang, 2022), and DreamGaussian (Tang et al., 2024) in three key aspects: 3D geometry, prompt consistency, and overall quality. We randomly select 96 prompts from the standard set of 153 prompts and generate 3D models, using Stable-DreamFusion (Tang, 2022), Latent-NeRF (Metzer et al., 2023), DreamGaussian (Tang et al., 2024), and our approach. Participants are then asked to rank the models based on the aforementioned criteria. As shown in Figure 5, our visual results outperform other methods across multiple metrics, attracting preferences from $56.25\%$ of participants for overall quality, $54.17\%$ for 3D geometry, and $47.92\%$ for prompt consistency. This highlights the efficacy of our approach across various evaluation criteria.

### 4.2 QUANTITATIVE EVALUATION

Since our task is a generation problem, we lack 3D ground-truth meshes for direct quantitative comparison of differences. Therefore, we follow the existing work, *i.e.*, DreamFusion (Poole et al., 2023), to evaluate the alignment between 2D projected images and the text prompt. In particular, we adopt the CLIP R-Precision (Radford et al., 2021) to evaluate the retrieval performance for both RGB images and depth maps. The RGB images serve as an indicator of texture quality, while the depth maps represent the geometric shape. A higher score indicates better performance. This evaluation is conducted using three pre-trained CLIP models with different model sizes, *i.e.*, CLIP B/32, CLIP B/16, and CLIP L/14. For a fair com-

Table 1: Quantitative comparisons with competitive methods. The best precision in every column is in **bold**. We do not include ProlificDreamer (Wang et al., 2023b) in this table, since it is extremely time-consuming, requiring about 11 hours per prompt for just the first training stage.

| Method | R-Precision (%) ↑ | | | | | |
|---|---|---|---|---|---|---|
| | CLIP B/32 | | CLIP B/16 | | CLIP L/14 | |
| | RGB | DEPTH | RGB | DEPTH | RGB | DEPTH |
| GT images | 77.1 | - | 79.1 | - | - | - |
| Latent-NeRF | 48.4 | 37.1 | 52.9 | 40.6 | 59.5 | 40.9 |
| Stable-Dreamfusion | 56.4 | 45.9 | 60.3 | 45.8 | 58.3 | 42.9 |
| DreamGaussian | 61.3 | 48.7 | 61.9 | 49.2 | 61.7 | 45.8 |
| **Ours** | **62.6** | **53.1** | **62.6** | **51.9** | **64.8** | **47.6** |

parison, we also adopt 153 standard prompts from Dreamfields (Jain et al., 2022). As shown in Table 1, we observe that our method consistently achieves the highest R-Precision scores across all three metrics in terms of both RGB texture and depth, indicating a significant advantage.

Table 2: The ablation study investigates the impact of different components, with the best precision highlighted in **bold** for each column. The ablation study validates the effectiveness of the proposed MTN architecture, progressive time step, and progressive radius. Notably, the full model (MTN-full) achieves the highest level of text-visual semantic alignment.

| Method | MTN | Progressive Time Step | Progressive Radius | R-Precision (%) ↑ | | | | | | | |
|---|---|---|---|---|---|---|---|---|---|---|---|
| | | | | CLIP B/32 | | CLIP B/16 | | CLIP L/14 | | Mean | |
| | | | | RGB | DEPTH | RGB | DEPTH | RGB | DEPTH | RGB | DEPTH |
| Single triplane | | | | 46.8 | 38.4 | 51.8 | 41.1 | 53.9 | 41.4 | 50.8 | 40.3 |
| MTN | ✓ | | | 57.8 | 46.7 | 58.2 | 46.2 | 62.2 | 42.8 | 59.4 | 45.2 |
| MTN-t | ✓ | ✓ | | 60.2 | 52.7 | 61.2 | 51.0 | 63.5 | 43.5 | 61.6 | 49.1 |
| MTN-r | ✓ | | ✓ | 57.9 | 48.5 | 60.4 | 48.8 | 62.4 | 42.7 | 60.2 | 46.7 |
| **MTN-full** | ✓ | ✓ | ✓ | **62.6** | **53.1** | **62.6** | **51.9** | **64.8** | **47.6** | **63.3** | **50.9** |

## 4.3 ABLATION STUDY AND FURTHER DISCUSSION

**Effectiveness of Multi-scale Triplanes.** We first investigate the impact of the multi-scale triplane architecture to substantiate its advantages. As shown in Table 2, we could observe that the multi-scale architecture facilitates both texture and geometric shape learning. Specifically, the RGB R-Precision is improved with a large margin $+8.6\%$ on average, while the mean depth R-Precision increases $+4.9\%$. We also provide a visualization result in Figure 6 (b). The basic single-scale triplane structure results in a 3D output that misses intricate details both texturally and geometrically, evident in incomplete hands, tails, and the presence of floating points. (Noted that for the single-triplane baseline, we use a resolution of $512 \times 512$, the same as the final resolution in our progressive multi-scale approach. This ensures that the single-triplane setup has comparable capacity to represent high-resolution details, allowing for a meaningful comparison.) In contrast, the multi-scale network gradually leverages the multi-scale information, yielding a more smooth geometric shape with clear edges. While there are still imperfections, the rabbit now possesses a more complete form, particularly noticeable in its overall silhouette.

**Effectiveness of Progressive Learning.** Here we further evaluate the impact of progressive time step sampling and progressive radius. (1) As shown in the third row of Table 2, the MTN with only progressive time step strategy could further improve the text alignment by $+2.2\%$ texture and $+3.9\%$ geometry quality on average. This is because the small time step towards the end of learning shifts the focus to high-frequency details, significantly improving the overall visual quality. (2) Similar to how humans often take a closer look to examine object details, our model, when applying the progressive radius approach, performs even better, showing a $+1.7\%$ improvement on the local texture details. As the camera gets closer, the 2D projection and the optimization objects both emphasize local quality, resulting in a refined 3D model. As a result, the culmination of these strategies leads to a final output that is both detailed and visually appealing (see Figure 6 (d)).

**Compatibility and Scalability.** The proposed method is compatible with various pre-trained diffusion models as supervision, and can be easily extended to further improve the quality of generation. For instance, our approach can integrate seamlessly with the state-of-the-art multi-view diffusion model MVDream (Shi et al., 2024), which effectively tackles the multi-face problem by emphasizing multi-view consistency. The combination enables a superior 3D consistency and exquisite textures and verifies the compatibility and scalability of our method (see Figure 7).

**3D Printing.** Our method provides a practical solution by directly converting the generated 3D output into a printable mesh format. The quality of these exported meshes is highlighted in Figure 10, which shows the uniformity of triangulation and the smoothness of surfaces. Such characteristics are crucial for the direct and efficient transmission of data to 3D printers. This is further evidenced by the six images on the right of Figure 10, showcasing the physical 3D products derived from these meshes. Our method significantly reduces the need for manual adjustments or additional post-processing steps, thereby streamlining the printing process.

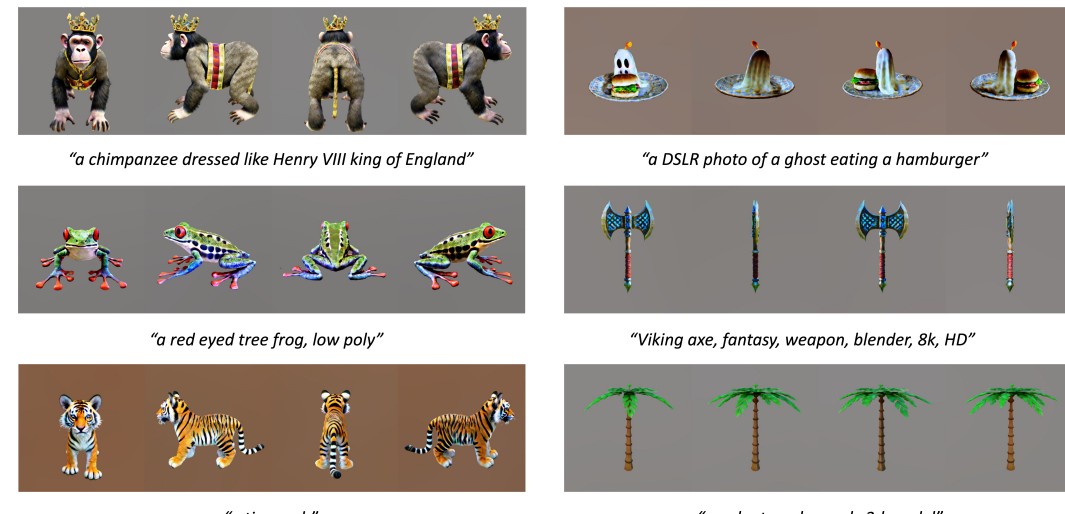

*"a chimpanzee dressed like Henry VIII king of England"*  *"a DSLR photo of a ghost eating a hamburger"*

*"a red eyed tree frog, low poly"*  *"Viking axe, fantasy, weapon, blender, 8k, HD"*

*"a tiger cub"*  *"a palm tree, low poly 3d model"*

Figure 7: Compatibility of the proposed method. Our method is highly compatible and can be easily scaled to other competitive multi-view diffusion models, such as MVDream (Shi et al., 2024), to further enhance the fidelity of 3D generation.

## 5 CONCLUSION

In this work, inspired by the bottom-up spirit, we introduce the Multi-scale Triplane Network (MTN) and a progressive learning strategy, both of which effectively ease the optimization difficulty during high-fidelity generation. The Multi-scale Triplane Network operates at the structure level to aggregate the multi-scale representation, while the progressive learning strategy functions at the recognition level to gradually refine high-frequency details. Extensive experiments verify the effectiveness of every component. We envision our approach offers a preliminary attempt for automatic 3D printing, bridging the gap between natural language descriptions and intricate 3D design. In the future, we will continue to explore the potential to complete occluded 3D objects (Mohammadi et al., 2023) via language prior and discriminative language guidance (Matsuzawa et al., 2023).

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

# A APPENDIX

## A.1 MORE RESULTS

**Qualitative Comparisons.** Figure 8 and 9 present additional qualitative comparisons, including recently introduced methods such as GSGEN and LucidDreamer, evaluated across a diverse set of prompts. As illustrated in these figures, our method consistently outperforms others in terms of visual quality, demonstrating sharper details and higher fidelity. Notably, both GSGEN and Lucid-Dreamer exhibit the multi-head (Janus) problem, leading to inconsistencies in multi-view rendering. Additional results can be found in the Supplemental Material.

Furthermore, Figure 10 provides a detailed comparison of the generated meshes rendered in Blender. Our method produces meshes with the most accurate geometric structure, achieving a high level of detail and realism. In contrast, other methods display significant distortions or geometric inaccuracies, further highlighting the robustness of our approach.

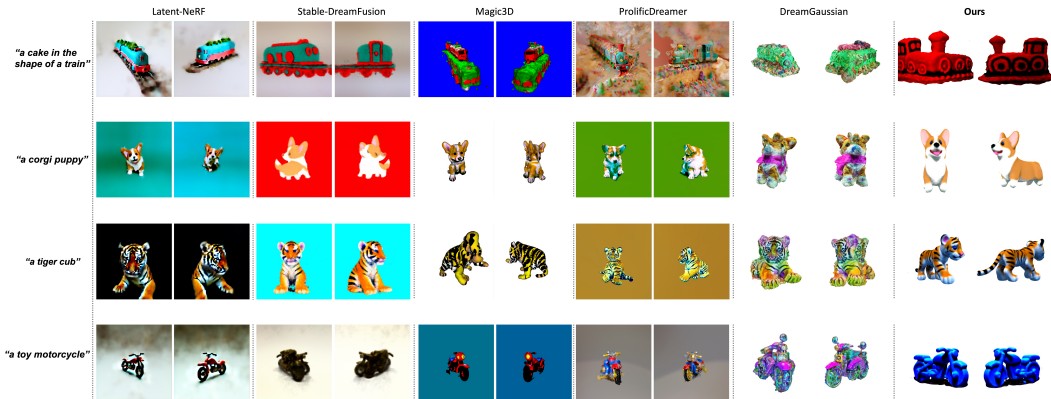

Figure 8: More qualitative comparisons.

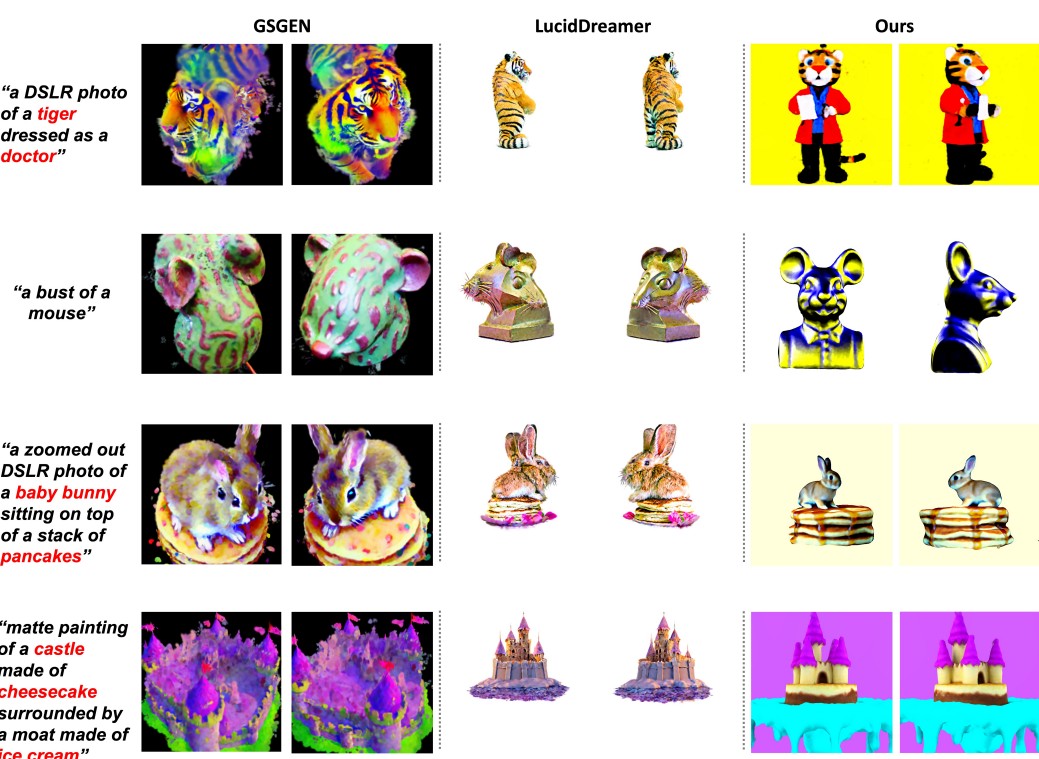

Figure 9: Qualitative comparisons with GSGEN and LucidDreamer.

Figure 10: Comparison of generated meshes (ready for print).

**Comparison with original MVDream.** We present the comparison between MTN + MVDream and the original MVDream in Table 3. Our MTN outperforms the NeRF component in MVDream, while requiring less training time and fewer parameters. It is important to note that we did not tune the hyperparameters for MTN, instead directly using those optimized for NeRF in the original MVDream.

Table 3: Comparison of NeRF backbone.

| NeRF Backbone | Diffusion | R-Precision (%) ↑ | Training Time ↓ | #Params↓ |
|---|---|---|---|---|
| NeRF (from MVDream) | MVDream | 67.1 | 1.5 hours | 12.6M |
| MTN (Ours) | MVDream | **67.9** | **1.3 hours** | **8.3M** |

**Time cost.** Details are presented in Table 4. All experiments are performed on a V100 GPU. Our model has the fastest convergence among NeRF-based methods. While Gaussian Splatting-based techniques converge quickly, they compromise on the quality of the generated results. Additionally, they require extra time for post-processing to make the generated objects ready for 3D printing.

Table 4: Comparison of methods (averaged on 153 prompts).

| Method | Type | R-Precision (%) ↑ | Training Time ↓ |
|---|---|---|---|
| Latent-NeRF | NeRF-based | 53.6 | ∼ 1 hour |
| Stable-DreamFusion | NeRF-based | 58.3 | ∼ 1.5 hours |
| Magic3D | NeRF-based | 62.0 | 2 hours |
| ProlificDreamer | NeRF-based | -* | > 20 hours |
| DreamGaussian | GS-based | 61.6 | 5 minutes |
| Ours | NeRF-based | **63.3** | ∼ 50 minutes |

*: Due to the limitation of GPU resources, ProlificDreamer (153 × 20 hours) precision is unavailable.

**Effectiveness of the Hierarchical Triplane.** Figure 11 (a) shows renderings with different combinations of triplanes. For example, the first image uses only the 64-triplane, while the second adds the 128-triplane, and so on. For the final triplane, we use a trivector to optimize memory usage and support higher resolutions. The total number of training iterations is the same for all four combinations in Figure 11 (a). This illustrates how the generation quality improves as higher-scale triplanes are added. Sampling from higher-resolution triplanes enhances details and sharp edges. More detailed explanations are provided in the "Why is a multi-scale structure crucial?" section in our paper.

**Effectiveness of Random Sampling in the final stage.** Unlike the concurrent DreamTime strategy (Huang et al., 2024), which does not use random sampling in the final stage, we adopt random sampling. As shown in Figure 11 (b), while keeping other hyperparameters consistent, our approach results in better illumination conditions and clearer edges compared to DreamTime (Huang et al., 2024).

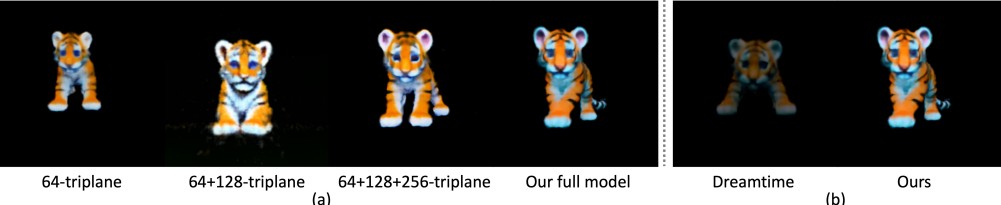

64-triplane    64+128-triplane    64+128+256-triplane    Our full model          Dreamtime          Ours
(a)                                                                              (b)

Figure 11: (a) Hierarchical outputs. (b) Ablation on strategies.

## A.2 Limitations

**Dependence on Diffusion Models:** The quality of the generated 3D outputs heavily depends on the pretrained diffusion model used for guidance. Limitations in the diffusion model's ability to interpret complex or ambiguous text prompts can propagate to our 3D generation results, occasionally leading to inaccuracies or oversimplified textures. A potential improvement is to incorporate fine-tuned diffusion models tailored for text-to-3D tasks or explore hybrid priors that combine text and geometry.

**Memory Constraints at Higher Resolutions:** While our multi-scale triplane architecture enables efficient optimization, scaling the resolution beyond 512 introduces significant GPU memory demands, making it challenging to train on consumer-grade hardware. This limits the applicability of our approach in scenarios requiring ultra-high resolution outputs. One possible solution is to adopt memory-efficient representations, such as compressed triplanes or mixed-resolution optimization strategies.

**Single-view SDS Framework:** Although our method is compatible with multi-view approaches (as demonstrated with MVDream in Figure 7), the experiments primarily focus on single-view SDS. This could lead to less robust multi-view consistency compared to methods specifically designed for multi-view supervision. Future work could incorporate multi-view consistency losses or explore integrating multi-view diffusion priors to enhance robustness.

