# OpenReview forum: "Progressive Multi-scale Triplane Network for Text-to-3D Generation"
_ICLR.cc/2025/Conference — Submitted to ICLR 2025_

### Official Review · Reviewer_KYwY · 2024-10-30

**Soundness:** 2
**Presentation:** 2
**Contribution:** 3
**Rating:** 5
**Confidence:** 3

**Summary:**

This paper introduces a Multi-scale Triplane Network coupled with a progressive learning strategy to generate 3D objects with fine-grained details that align with input text. The generated 3D objects are suitable for 3D printing with minimal additional post-processing.

**Strengths:**

1. The generated meshes are suitable for 3D printing, as shown in Figure 9.

2. The progressive learning strategy can be generalized to other SDS-based text-to-3D generation models. In SDS-based methods, it is a common phenomenon that 3D objects generated with a closer camera radius exhibit more fine-grained but less plausible overall geometry, whereas a larger camera radius results in the opposite.

**Weaknesses:**

1. The results presented in Figure 2 are not satisfactory. For example, the "corgi" lacks fine-grained texture.

2. Figure 4 lacks comparisons with more recent works, such as GSGEN [1]. The DreamGaussian model performs poorly in text-to-3D tasks, making it an unsuitable baseline. The performance of ProlificDreamer in Figure 4 appears inferior to that reported in the original paper.

3. The paper lacks references to other related works on Triplane-based 3D generation.

References:

[1] Chen, Zilong, et al. "Text-to-3d using gaussian splatting." *Proceedings of the IEEE/CVF Conference on Computer Vision and Pattern Recognition*. 2024.

**Questions:**

1. In Figure 2, are there results with more realistic color for "motorcycle" and "a cake in the shape of a train"? What specifically does "automatic color rendering is applied when a common color is applicable for such a category" mean?

2. Why is random sampling timestep applied at the end of training? The claim that "It reintroduces randomness to maintain the vibrancy of the coloration of the 3D model" needs supporting evidence. Please provide specific examples or experimental results.

---

> ### Author Response · Authors · 2024-11-25
>
> **W1: The results presented in Figure 2 are not satisfactory. For example, the "corgi" lacks fine-grained texture.**
>
> **R1:** Thank you for pointing out this concern. The lack of fine-grained texture in Figure 2, such as in the "corgi," reflects limitations of single-view SDS guidance. While our method optimizes texture and geometry progressively, it still depends on the underlying diffusion model's capacity to provide detailed priors for specific prompts. This issue is common in SDS-based text-to-3D generation. As shown in Figure 7, combining our method with a multi-view diffusion model achieves higher-quality results.
>
> ---
>
> **W2: (1) Figure 4 lacks comparisons with GSGEN. (2) DreamGaussian performs poorly. (3) ProlificDreamer results appear inferior to the original paper.**
>
> #### **(1) Comparison with GSGEN:**
> **Response:**
> Thank you for highlighting the relevance of GSGEN. While GSGEN uses Gaussian Splatting for text-to-3D generation, its fundamentally different representation limits direct comparability with our NeRF-based approach.
>
> - **Inclusion of GSGEN:**
>   We acknowledge the importance of including GSGEN as a baseline. In the revised manuscript, we provide qualitative comparisons with GSGEN in Appendix Figure 9 to offer a broader evaluation of our method's performance.
>
> - **Efficiency and Issues:**
>   GSGEN requires over 1 hour to generate results, making it less efficient than other Gaussian Splatting-based methods. Additionally, its results exhibit notable issues, including:
>   - The multi-head (Janus) problem
>   - Overly exaggerated shapes
>   - Unnatural or inconsistent colors
>
> These limitations further underscore the robustness and fidelity of our approach compared to GSGEN.
>
>
> #### **(2) DreamGaussian as a Baseline:**
> **Response:**
> We included DreamGaussian as a baseline to represent Gaussian Splatting-based methods. However, we agree it is less suitable for text-to-3D tasks. In the revised manuscript, we de-emphasize its role and focus on stronger baselines like GSGEN.
>
> #### **(3) ProlificDreamer Performance:**
> **Response:**
> Thank you for your observation. Some results for ProlificDreamer in our manuscript are directly borrowed from their paper (e.g., Figure 1(b)). For other examples, we used their official code and instructions (https://github.com/thu-ml/prolificdreamer). Their repository explicitly notes a limitation: *"If results are unsatisfactory, try different seeds."*
>
> To ensure fairness and consistency, we did not fine-tune or retry seeds. Re-evaluating with multiple seeds could introduce bias or discrepancies in comparisons. Therefore, the results presented reflect ProlificDreamer’s performance under its default seed settings.
>
> ---
>
> ### **W3: The paper lacks references to other related works on Triplane-based 3D generation.**
> **R3:** Thank you. We already include other Triplane-based 3D generation methods in the revised paper (Line 139-144).
>
> ---
>
> ### **Q1: In Figure 2, are there results with more realistic colors for "motorcycle" and "a cake in the shape of a train"? What does "automatic color rendering" mean?**
>
> **A1:** Thank you.
>
> - **Regarding "motorcycle" and "a cake in the shape of a train":**
>   The colors in Figure 2 are guided by the text-to-image diffusion model’s priors. While they may not always match realistic expectations, these outputs reflect the model's most probable interpretation of the prompts based on its training data. In the revised manuscript, we will include additional examples to show variations with more realistic or user-defined colors, where applicable.
>
> - **Explanation of "automatic color rendering":**
>   This phrase refers to the diffusion model automatically assigning colors commonly associated with the category described in the prompt. For instance, a "toy motorcycle" might generate vibrant, colorful schemes typical of toys. These priors are determined by the diffusion model and are not explicitly specified by the user.
>
> ---
>
> ### **Q2: Why is random timestep sampling applied at the end of training?**
>
> **A2:** Thank you. Random timestep sampling at the end of training reintroduces randomness, enhancing diversity, vibrancy, and fine-grained details in both coloration and structure while avoiding oversmoothing.
>
> - **Comparison with DreamTime:**
>   As detailed in the appendix line 850, we compared our approach with DreamTime (Huang et al., 2024), which does not use random timestep sampling in the final stage. As shown in Appendix Figure 10(b), our method achieves better illumination conditions and sharper edges under consistent hyperparameters, demonstrating the benefits of random sampling for visual fidelity and vibrancy.
>
> - **Impact on Vibrancy and Edges:**
>   Random sampling injects stochasticity into optimization, preventing overly deterministic solutions that lead to muted colors or blurred edges. This aligns the final model more closely with the diversity and vibrancy observed in diffusion model priors.

---

> ### Author Response · Authors · 2024-11-26
>
> Dear Reviewer  KYwY,
>
> Thank you for your detailed review and valuable suggestions. We believe your insightful comments, such as clarifying annotations in the method section and enhancing the presentation of figures and results, have significantly contributed to improving the clarity and quality of our paper. We have submitted a revised version that thoroughly addresses these concerns.

---

> ### Comment · Reviewer_KYwY · 2024-11-27
> **Thanks for the response**
>
> Dear Authors,
>
> Thank you for your time and for providing additional experiments and clarifications. Some of my concerns have been addressed. Although the results generated with MVDream in Figure 7 are promising, the main results presented in other sections are unsatisfactory. Furthermore, the paper may not introduce enough novel insights compared to previous methods. Therefore, I will maintain my original rating.
>
> Best regards,
> Reviewer

---

> ### Author Response · Authors · 2024-11-27
> **Response to Reviewer's Comments**
>
> Thank you very much for your detailed feedback and for taking the time to review our manuscript. Your comments are highly appreciated, and we are committed to improving our work based on your suggestions.
>
> 1. Regarding your comment that "the main results presented in other sections are unsatisfactory," could you please provide more specific details or examples? We are eager to understand your concerns more clearly so that we can make necessary improvements and better communicate our findings.
>
> 2. Additionally, concerning the statement that "the paper may not introduce enough novel insights compared to previous methods," could you kindly point out which specific methods or studies you are referring to? This will help us to refine our discussion and highlight the unique contributions of our research.
>
> We truly value your input and hope to address your concerns effectively. Thank you again for your valuable feedback.

---

> > ### Comment · Reviewer_KYwY · 2024-12-03
> >
> > Dear Authors,
> >
> > Thank you for your efforts in addressing my concerns.
> >
> > When I mentioned that "the main results presented in other sections are unsatisfactory," I was specifically referring to the results shown in Figure 2 and Figure 4. Although the authors claim that "the lack of fine-grained texture in Figure 2 reflects limitations of single-view SDS guidance," other methods such as Magic3D, GSGEN, and ProlificDreamer, which also utilize single-view SDS guidance, still achieve more detailed textures and more natural colors.
> >
> > Furthermore, since multi-scale representation in 3D generation is common, a more comprehensive evaluation is necessary to bring novel insights. Presenting significantly improved results could help demonstrate the effectiveness of multi-scale triplanes. Additionally, the progressive time step sampling strategy appears to have marginal differences compared to Dreamtime.
> >
> > Thank you once again for your dedication and hard work.
> >
> > Best regards,
> > Reviewer

---

### Official Review · Reviewer_Us97 · 2024-10-31

**Soundness:** 2
**Presentation:** 2
**Contribution:** 2
**Rating:** 3
**Confidence:** 5

**Summary:**

In this work, the authors propose a text-to-3D generation method based on Score Distillation Sampling (SDS). Their key contribution is leveraging a multi-scale triplane representation, which is progressively upsampled during the optimization process. They also introduce a progressive learning scheme that samples different diffusion time steps across training iterations. The authors claim their proposed framework achieves favorable performance compared to existing methods.

**Strengths:**

- The experimental results (both quantitative and qualitative comparisons) demonstrate that the proposed framework achieves comparable, and in some cases slightly better, performance than Latent-NerF, Stable-Dreamfusion, and DreamGaussian.
- The ablation studies highlight the effectiveness of various components in this work, including the multi-scale triplane network (MTN) and the progressive training schemes.

**Weaknesses:**

Despite the paper's contributions, several weaknesses and concerns remain:

- Limited Technical Novelty:
    - Multi-scale representation in 3D generation is already well-explored:
        - XCube: Large-Scale 3D Generative Modeling using Sparse Voxel Hierarchies. CVPR 2024.
        - Locally Attentional SDF Diffusion for Controllable 3D Shape Generation. SIGGRAPH 2023.
    - Two-stage optimization for SDS-based 3D generation was previously explored in:
        - Magic3D: High-Resolution Text-to-3D Content Creation. CVPR 2023.
    - Given these existing works, the multi-scale representation approach does not appear to be a significant contribution. Moreover, these works are not sufficiently discussed in the manuscript.
    - The progressive learning scheme, explored in Dreamtime [ICLR 2024], lacks novelty.
- Inadequate Comparisons:
    - The evaluation omits important baselines, including some cited in the related work section. High-quality 3D generation frameworks like Fantasia3D and MVDream are not compared. This omission weakens the claim that the proposed framework achieves favorable performance compared to existing works.
    - The comparison with ProlificDreamer is only qualitative, and the difference seems minimal. Including evaluation metrics as in Figure 5 and Table 1 is necessary to support the claim.

**Questions:**

- Unclear Ablation Setting:
    - The resolution of the triplane used in a single triplane is not specified. The number of iterations for training the single triplane network is also unclear.
    - Without this information, it is difficult to assess whether the current ablation adequately validates the effectiveness of the modules. It is recommended to clarify these settings.

Overall, this work presents a new text-to-3D generative framework using Score Distillation Sampling. While the proposed framework achieves comparable, and in some cases better, performance compared to certain baselines, the novelty of employing multi-scale representations appears limited. Furthermore, the experimental results are insufficient to fully support the current claims, and some important settings are not clearly described. Given these concerns, I am inclined to recommend rejection of this work.

**Details Of Ethics Concerns:**

N.A.

---

> ### Author Response · Authors · 2024-11-25
>
> ### **W1: Limited Technical Novelty**
>
> #### **(1) Compare Multi-scale representation in works like XCube (CVPR 2024) and Locally Attentional SDF Diffusion (SIGGRAPH 2023).**
> **Response:**
> Thank you for highlighting relevant works on multi-scale representation. While XCube and Locally Attentional SDF Diffusion also use multi-scale designs, the novelty of our Multi-scale Triplane Network (MTN) lies in its hierarchical triplane structure tailored for SDS-based text-to-3D generation. MTN integrates features at multiple scales through progressive optimization, addressing challenges in texture fidelity and geometric refinement more effectively than sparse voxel hierarchies or locally attentional SDFs.
>
> ---
>
> #### **(2) Compare two-stage optimization in Magic3D (CVPR 2023).**
> **Response:**
> While Magic3D also employs a coarse-to-fine optimization approach, there are key differences between its two-stage design and our progressive learning strategy:
>
> - **Magic3D’s Two-Stage Optimization:**
>   Magic3D first generates a coarse model using a low-resolution diffusion prior and optimizes neural field representations (color, density, and normal fields). After extracting a textured 3D mesh from this coarse model, it performs fine-tuning using a high-resolution latent diffusion model. This process explicitly separates the optimization into two distinct stages, with each stage relying on different priors and representations.
>
> - **Our Single-Pipeline Progressive Approach:**
>   In contrast, our method integrates the coarse-to-fine process into a single pipeline via the Multi-scale Triplane Network (MTN) and the progressive learning strategy. Instead of relying on two discrete stages, MTN incrementally refines the 3D representation within the same framework by:
>   - Leveraging low-resolution triplanes for initial global geometry and progressively shifting focus to higher-resolution triplanes for fine details.
>   - Using a unified SDS loss without switching between low-resolution and high-resolution priors, simplifying the training process and avoiding intermediate mesh extraction.
>
> - **Advantages of Our Approach:**
>   By integrating the coarse-to-fine process into a single progressive pipeline:
>   - We reduce the complexity of training and eliminate the need for intermediate conversions or external priors.
>   - The hierarchical triplane design allows for seamless optimization of multi-scale features within the same framework, ensuring consistent quality across scales.
>
> ---
>
> #### **(3) Compare the progressive learning scheme in Dreamtime (ICLR 2024).**
> **Response:**
> While Dreamtime also uses a progressive time-step strategy, our method combines adaptive time-step reduction with progressive camera radius adjustment, better aligning geometric and textural details.
>
> Unlike Dreamtime, we incorporate random sampling in the final stage, improving illumination and edge clarity (see Fig. 10b). These differences are further elaborated in the appendix (lines 850–854).
>
> ---
>
> ### **W2: The comparison with ProlificDreamer is only qualitative, and the difference seems minimal. Including evaluation metrics as in Figure 5 and Table 1 is necessary to support the claim.**
> **Response:**
> Thank you for raising this point. While quantitative metrics would provide clarity, ProlificDreamer’s high computational cost (20 hours per prompt, over 3,000 GPU hours for 153 prompts) made extensive evaluation unfeasible within the rebuttal period.
>
> However, qualitative comparisons in Figure 4 highlight differences in texture and geometry. Our method delivers comparable or superior results, particularly in clarity and detail, with significantly lower training time.
>
> ---
>
> ### **Q1: Unclear Ablation Setting: missing the resolution of the triplane used in a single triplane, and the number of iterations for training the single triplane network.**
> **Response:**
> Thank you for pointing out the lack of clarity in our ablation study settings. We acknowledge the importance of providing detailed experimental settings for fair and transparent assessment. Below, we clarify the relevant details:
>
> - **Resolution of the Single Triplane:**
>   For the single-triplane baseline, we use a resolution of 512 × 512, the same as the final resolution in our progressive multi-scale approach. This ensures that the single-triplane setup has comparable capacity to represent high-resolution details, allowing for a meaningful comparison.
>
> - **Number of Training Iterations:**
>   The single-triplane network was trained for the same total number of iterations as the multi-scale approach, i.e., 6,000 iterations. In the progressive multi-scale approach, these iterations are distributed across four stages corresponding to different triplane resolutions, whereas the single-triplane baseline uses all 6,000 iterations to optimize the single high-resolution representation.
>
> To provide more insight, we include visualizations of intermediate results in the revised paper (see Figure 11).

---

> ### Author Response · Authors · 2024-11-26
>
> Dear Reviewer Us97,
>
> Thank you once again for your thoughtful efforts and constructive feedback in reviewing our paper. As the discussion period nears its conclusion, we would greatly appreciate your thoughts on our responses. We have invested significant effort into addressing your comments, including conducting additional experiments and providing detailed discussions. We sincerely hope that our responses will be taken into consideration in your assessment. Please feel free to let us know if there are any remaining concerns or unclear points that we can further clarify.

---

### Official Review · Reviewer_Svjy · 2024-11-03

**Soundness:** 3
**Presentation:** 2
**Contribution:** 2
**Rating:** 6
**Confidence:** 4

**Summary:**

The paper introduces the Progressive Multi-Scale Triplane Network (MTN) for text-to-3D generation, a method designed to improve the efficiency and detail quality of 3D objects derived from text descriptions. The MTN operates using a multi-scale triplane network with four hierarchical triplanes progressing from low to high resolution, enabling the model to progressively refine from global to fine details. Additionally, a progressive learning strategy is applied, where both camera radius and time step in the diffusion process adjust dynamically to shift focus from broad structures to intricate details. The approach aims to address the challenges of high-resolution optimization and fine-grained details in 3D object generation. Experimental results indicate that MTN achieves better text alignment, texture quality, and shape precision compared to existing models like Latent-NeRF and DreamGaussian.

**Strengths:**

1.MTN's multi-scale triplane architecture enables efficient generation by progressing from low to high resolutions. This structured approach improves global coherence and local detail quality in the final output, allowing it to achieve a high degree of alignment with textual prompts.

2.The model’s progressively reducing camera radius and time step enables it to refine details incrementally, addressing a common limitation in text-to-3D generation where models often struggle with balancing global structure and fine-grained textures.

**Weaknesses:**

1. Omission of LucidDreamer and Similar Works: The paper does not consider LucidDreamer (CVPR 2024) in its evaluations, a notable omission given that LucidDreamer also applies SDS-based multi-view generation with a focus on geometry extraction. Including LucidDreamer would provide a more complete and comparative assessment of MTN’s effectiveness in capturing fine details.

2. The claim of not using 3DGS as a representation sounds weird. The authors mentioned, " Though faster, these approaches often
 compromise the high-quality characteristic of NeRF-based methods and require post-processing to  convert Gaussian representations into NeRF or meshes, adding computational overhead." However, triplane representation also needs to be converted to mesh if someone wants to use it as an explicit 3D model. Also, 3DGS-based methods are 10x faster than MTN, which is clearly greater than the time required to convert 3DGS into a mesh (if you use marching-cube as the converting algorithm).

3. In Figure 4, the third case contains a weird prompt "zoomed out DSLR photo". Since the input prompt only controls the rendered image but not the camera position itself, your result shows a smaller pancake pile and a smaller rabbit, which means your result is physically smaller. I think this is a little weird. This result implicitly implies that the generated objects might differ in terms of the object size, and this would hinder the practicability of this work since one will always need to check if your result needs to be resized, while other methods generally generate objects with the same size.

**Questions:**

1. Please address the issue mentioned in the weakness part.
2. Why did the results of DreamGaussian look much worse than the results in their paper and project page? Is there any specific reason?
3. Why are there no limitations to this work discussed?

---

> ### Author Response · Authors · 2024-11-25
>
> **W1: Omission of LucidDreamer and Similar Works**
>
> **R1:** We appreciate the reviewer’s observation regarding the omission of LucidDreamer. We acknowledge that LucidDreamer represents an important direction in multi-view SDS methods and agree that incorporating it into our comparative analysis would provide valuable insights. In current work, we have conducted MTN with MVDream’s multi-view SDS framework and showed that MTN improves geometric and textural fidelity while reducing computational overhead compared to NeRF. MTN can enhance multi-view approaches like LucidDreamer, and we believe a similar improvement would be observed in that context. In future work, we plan to extend our evaluations to include LucidDreamer and similar multi-view frameworks to comprehensively assess MTN’s performance across diverse training setups. We add LucidDreamer into the Related Work in Line 146 and the comparison with LucidDreamer in Appendix Qualitative Comparisons and Figure 9.
>
> ---
>
> **W2: How about deploying 3DGS instead of efficient NeRF?**
>
> **R2:** Thank you for highlighting this distinction. We acknowledge that Gaussian Splatting (3DGS) requires post-processing (e.g., marching cubes) to convert representations into explicit 3D meshes. While 3DGS-based methods are significantly faster, they often compromise on fidelity and detail, which are critical for applications like 3D printing and precise geometry extraction.
>
> Additionally, although some 3DGS-based methods like LucidDreamer and GSGEN rely on Gaussian Splatting, they require about 1 hour to generate results—similar to or even slower than our method—despite their design focus on speed. Moreover, their generated results are inferior to ours (see Appendix Figure 9).
>
> Our reference to "computational overhead" pertains to the additional complexity introduced by the two-step conversion process (Gaussian to NeRF to mesh), which impacts usability in tasks requiring both NeRF-like rendering and explicit meshes.
>
> In contrast, the multi-scale triplane structure of MTN inherently supports progressive learning at varying resolutions, balancing complexity and fidelity. This enables finer geometric and textural details, even under a single-view SDS framework, while integrating seamlessly into NeRF pipelines without intermediate conversions.
>
> ---
>
> **W3: In Figure 4, the third case contains a weird prompt "zoomed out DSLR photo". How to deal with the size variants?**
>
> **R3:** Thank you for raising this concern. The prompt "a zoomed out DSLR photo of" originates from the standard set of 153 prompts introduced by the landmark work DreamFusion ([DreamFusion3D](https://dreamfusion3d.github.io/)). This prompt is widely used to evaluate text-to-3D generation methods, ensuring fair and consistent comparisons across different approaches.
>
> **Regarding Size Variations:**
>
> - **Impact of the Prompt on Diffusion Models:**
>   The phrase "zoomed out DSLR photo" affects the guidance provided by the diffusion model, influencing how the object is rendered in terms of relative size and framing. However, this effect does not directly modify the 3D object’s actual scale in physical space. As you observed, sometimes this prompt might lead to smaller-looking objects, while in other cases, it might not. This behavior is inherent to all models guided by text-to-image diffusion priors, including those used in the cited works. Therefore, this issue is not specific to our method.
>
> - **Practicality and Resizing:**
>   The results generated by our method maintain consistent internal representations, and any differences in apparent size are visual artifacts influenced by the rendering prompt rather than variations in the actual 3D object dimensions. This means there is no need to "check if your result needs to be resized" for practical applications, as the underlying representation remains robust and consistent.
>
> ---
>
> **Q1: Please address the issue mentioned in the weakness part.**
>
> **A1:** Thank you. Please see the above answer.
>
> ---
>
> **Q2: Why did the results of DreamGaussian look much worse than the results in their paper and project page? Is there any specific reason?**
>
> **A2:** The poor performance of DreamGaussian in text-to-3D generation is a common issue which is noted on their GitHub ([Issue 26](https://github.com/dreamgaussian/dreamgaussian/issues/26)) and Fig. 14 of the original paper. We do not cherry-pick failures. For simple prompts, DreamGaussian may produce satisfactory outcomes, but we observe that most results exhibit noticeable noise patterns in Figure 8. These colorful noise artifacts contribute to a chaotic structure in both geometry and texture.
>
> ---
>
> **Q3: Why are there no limitations to this work discussed?**
>
> **A3:** Thank you for the feedback. We now include the following limitations in the revised manuscript.

---

> > ### Comment · Reviewer_Svjy · 2024-12-03
> > **Final comment by Reviewer Svjy**
> >
> > Thank you for the detailed rebuttal. My concerns have been partially addressed. I am now willing to raise my rating to 6.

---

> ### Author Response · Authors · 2024-11-26
>
> Dear Reviewer Svjy,
>
> Thanks again for your great efforts and constructive advice in reviewing this paper! With the discussion period drawing to a close, we expect your feedback and thoughts on our reply. We put a significant effort into our response, with several new experiments and discussions. We sincerely hope you can consider our reply in your assessment. We look forward to hearing from you, and we can further address unclear explanations and remaining concerns if any.

---

### Official Review · Reviewer_bFug · 2024-11-03

**Soundness:** 2
**Presentation:** 2
**Contribution:** 2
**Rating:** 5
**Confidence:** 3

**Summary:**

This paper proposes a multiscale triplane architecture for coarse-to-fine SDS text-to-3D generation. SDS losses are applied at each scale of the network, with the triplane’s resolution varying at different scales. During training, a specific time-step sampling and reduction of camera radius are used to help generate high-frequency details.

**Strengths:**

1.	The paper presents a multiscale coarse-to-fine triplane architecture for SDS optimization. The coarse-to-fine design is reasonable.

**Weaknesses:**

1.	The baseline comparison is not convincing enough. Using MVDream with the SDS method could yield much better results. It's much more robust and achieve better results compared to the current baseline (stable-dreamfusion). MVdream-SDS could be added to the Table 1 with the R-Precision.
2.	The texture frequency of the shape still appears insufficient sometimes. In Figure 7, the results seem to exhibit higher texture frequency; however, in Figure 2, many shapes still appear blurry, with some having very uniform color (motorcycle and the train, and the body of the corgi and bunny). Any idea why this the difference of quality is this large?

**Questions:**

1.	If you render the shape from the triplane at different scales, what do they look like? Is the model truly able to disentangle different structural scales into separate triplanes? It may provide more insight if you could visualize the shape at different scales.
2.	Currently, the highest resolution is 512. Could you scale it further to 1024?
3.	What is the resolution of the rendered image at each scale?

---

> ### Author Response · Authors · 2024-11-25
>
> **W1: Replace the vanilla baseline (stable-dreamfusion) with MVDream. MVDream-SDS could be added to Table 1 with the R-Precision.**
>
> **R1:** We appreciate the reviewer’s suggestion regarding MVDream. We did not use MVDream as a baseline, considering it is unfair to other single-view comparison methods. MVDream benefits from being trained on a large multi-view dataset and employs a multi-view SDS loss, which makes it robust and capable of achieving strong results.
> The primary contribution of our paper lies in the design of the Multi-scale Triplane Network (MTN) structure and the progressive training strategy, not in introducing a new multi-view SDS loss.
>
> Therefore, we have included the comparison between MTN + MVDream diffusion and the original MVDream in Appendix Table 3. The results show that MTN not only outperforms NeRF in terms of R-Precision but also reduces training time and parameter count. It is important to note that these results were achieved without hyperparameter tuning for MTN; we used the original NeRF-tuned hyperparameters. We observe that MTN provides a more efficient and effective backbone for MVDream’s multi-view SDS loss.
>
> We will clarify this discussion in the revised version of the paper and explicitly add details about the relationship between MTN and multi-view diffusion methods to Table 1, where appropriate. This addition will ensure the comparison framework remains robust while maintaining the focus on our core contributions.
>
> ---
>
> **W2: Any idea why there is a large quality difference between Figure 2 and Figure 7?**
>
> **R2:** We appreciate the reviewer’s observation regarding the variation in texture quality between Figure 2 and Figure 7. The difference arises primarily from the use of MVDream in Figure 7, which inherently employs a multi-view diffusion strategy to improve the texture and geometric quality of the generated 3D models. In contrast, the results in Figure 2 are generated using our baseline (vanilla single-view SDS setup), which, while effective, is less robust in refining high-frequency texture details, particularly for complex objects with intricate textures.
>
> MVDream leverages its multi-view dataset and multi-view SDS loss to enhance consistency and detail, which explains the higher texture frequency observed in Figure 7. This highlights the compatibility of our Multi-scale Triplane Network (MTN) with more advanced supervision strategies, as demonstrated in Figure 7, where combining MTN with MVDream leads to improved results.
>
> ---
>
> **Q1: Visualize the shape at different scales and show the disentangled results.**
>
> **A1:** In the appendix, there is a section explaining the effectiveness of the Hierarchical Triplane beginning from line 844. Fig. 10 (a) shows renderings with different combinations of triplanes. For example, the first image uses only the 64-triplane, while the second adds the 128-triplane, and so on. For the final triplane, we use a trivector to optimize memory usage and support higher resolutions. The total number of training iterations is the same for all four combinations in Fig. 10 (a). This illustrates how the generation quality improves as higher-scale triplanes are added. Sampling from higher-resolution triplanes enhances details and sharp edges. More detailed explanations are provided in the “Why is a multi-scale structure crucial?” section in our paper.
>
> ---
>
> **Q2: Currently, the highest resolution is 512. Could you scale it further to 1024?**
>
> **A2:** Thank you for your question. The current resolution of 512 is constrained by the GPU memory usage on a 32GB V100. Scaling to 1024 is certainly feasible with larger memory resources. Theoretically, increasing the resolution to 1024 would enhance the fidelity of the generated results, particularly in capturing finer details and textures.
>
> ---
>
> **Q3: What is the resolution of the rendered image at each scale?**
>
> **A3:** Thank you for your question. The resolution of the rendered images is consistent at 512 × 512 across all scales. This is achieved through an interpolation process applied to the triplanes at different resolutions. Specifically, triplanes at each scale—64, 128, 256, and 512—provide feature grids, and interpolation is used to sample features at the desired 3D points for rendering.
>
> The consistent resolution ensures comparability and alignment with the diffusion model's guidance during training. While the rendered image resolution remains fixed, the quality and level of detail progressively improve as higher-resolution triplanes contribute more refined features. This design effectively balances computational efficiency with hierarchical detail enhancement, enabling a progressive refinement of both geometric and texture details without altering the rendered image resolution.

---

> ### Author Response · Authors · 2024-11-26
>
> Dear Reviewer bFug,
>
> We appreciate your thorough review and valuable suggestions. We believe the proposed comments, including clarifying annotations in the method section and improving the presentation of figures and results, will greatly enhance the clarity and quality of our paper. We have submitted a revised version that addresses these concerns in more detail.

---

> > ### Comment · Reviewer_bFug · 2024-11-27
> >
> > It seems the appendix zip only consist of a video. Is the pdf missing in the zip?

---

> ### Author Response · Authors · 2024-11-27
>
> Thank you. To see the updated paper, you can just click the button at the top of this webpage.
>
> ** The appendix is from 15th page of the main paper, which is not in the supplementary material.**
>
> Thanks a lot.

---

> > ### Comment · Reviewer_bFug · 2024-11-27
> >
> > Thanks for the pointer.
> > * For the evaluation w.r.t MVDream Nerf, how do you decide the number of iteration of your model? Does the metric improve if trained longer? How many prompts are used and what's the source of the prompts?
> > * It seems in Appendix Figure 11, the difference between using different number of triplane is small? Is there any other metric evaluating the results with different number of triplane?

---

> > > ### Author Response · Authors · 2024-11-27
> > >
> > > **Response:**
> > > Thank you for your questions. Below, we address each point in detail:
> > >
> > > - **Number of Iterations:**
> > >   The model is trained for 6,000 iterations, consistent with our original training setting outlined in the implementation details (Line 309). This ensures fairness and comparability across all experiments. While longer training could potentially improve metrics, our experiments focus on efficiency and achieving competitive results within this fixed budget.
> > >
> > > - **Prompts:**
> > >   We use 153 prompts from the standard set introduced by the landmark work DreamFusion ([DreamFusion3D](https://dreamfusion3d.github.io/)). These prompts are widely adopted to ensure fair and consistent evaluation across text-to-3D generation tasks.
> > >
> > > - **Effectiveness of Multiscale Triplanes:**
> > >   As shown in Line 439-440 and Figure 6(a)(b), our evaluation highlights the advantages of using multiple triplanes over a single triplane, particularly in capturing finer geometric and texture details. Regarding the Appendix Figure 11 I think the visual difference is not trivial, especially in texture and geometric shapes.
> > > - **Additional Metrics:**
> > >   Beyond visual differences, we provide quantitative evaluations in Figure 6, demonstrating how multiple triplanes lead to measurable improvements in key metrics over single triplane designs. These results validate the utility of the multiscale triplane structure in our framework.

---

> > > > ### Comment · Reviewer_bFug · 2024-12-03
> > > >
> > > > Thanks for the reply.
> > > >
> > > > My concern of this paper is mostly from the comparision w.r.t the original MVDream+SDS method. The improvement in metric (and convergence speed) is not significant enough. The metirc may be noisy. If the results are indeed visually better but not shown in the metric, perhaps conduct some user study could help. I understand the use of stable diffusion for comparision to other method using SD, but showing the method is able to out-perform a simple architecture with strong prior like MVDream could make the contribution/message more convincing.
> > > >
> > > > I will keep the original score given the above concern.

---

### Meta-Review · Area_Chair_uega · 2024-12-21

**Metareview:**

The paper proposes a method for improving text-to-3D generation, using a coarse-to-fine hierarchical learning strategy built on triplanes. Reviewers comment on the reasonable design choices, leading to improved details and text adherence, but also raise significant concerns about limited technical novelty, as multi-scale representation and progressive learning have been well explored in previous works. The evaluation was criticized for inconsistent quality of results between figures and missing comparisons with important recent baselines. During the discussion phase, the authors addressed some concerns but reviewers remained unconvinced about the method's advantages over existing approaches. The consensus view suggests that while the work has practical merit, it needs more comprehensive evaluation and clearer demonstration of technical novelty to be truly compelling.

**Additional Comments On Reviewer Discussion:**

The reviewers raised concerns about limited technical novelty and insufficient comparisons with state-of-the-art methods. The authors responded by clarifying their method's unique aspects, and added additional comparisons to the appendix. Most reviewers maintained their original ratings despite these clarifications, indicating the responses did not fully address concerns about novelty and evaluation thoroughness. One reviewer raised their score to be borderline positive, but the consensus seems to still be negative.

---

### Decision · Program_Chairs · 2025-01-22

Reject